# Where Are Smart Cities Heading? A Meta-Review and Guidelines for Future Research

João Reis *, Pedro Alexandre Marques and Pedro Carmona Marques

Industrial Engineering and Management, Faculty of Engineering, Lusofona University and EIGeS, Campo Grande, 1749-024 Lisbon, Portugal
* Correspondence: joao.reis@ulusofona.pt

**Abstract:** (1) Background: Smart cities have been gaining attention in the community, both among researchers and professionals. Although this field of study is gaining some maturity, no academic manuscript yet offers a unique holistic view of the phenomenon. In fact, the existing systematic reviews make it possible to gather solid and relevant knowledge, but still dispersed; (2) Method: through a meta-review it was possible to provide a set of data, which allows the dissemination of the main theoretical and managerial contributions to enthusiasts and critics of the area; (3) Results: this research identified the most relevant topics for smart cities, namely, smart city dimensions, digital transformation, sustainability and resilience. In addition, this research emphasizes that the natural sciences have dominated scientific production, with greater attention being paid to megacities of developed nations. Recent empirical research also suggests that it is crucial to overcome key cybersecurity and privacy challenges in smart cities; (4) Conclusions: research on smart cities can be performed as multidisciplinary studies of small and medium-sized cities in developed or underdeveloped countries. Furthermore, future research should highlight the role played by cybersecurity in the development of smart cities and analyze the impact of smart city development on the link between the city and its stakeholders.

**Keywords:** smart cities; meta-review; smart city dimensions; digital transformation; sustainability; resilience; megacities; cybersecurity; privacy

---



## 1. Introduction

In recent years, smart city initiatives and projects have grown around the world [1]. This progress has been in part driven by recent advances in Artificial Intelligence (AI) and the Internet of Things (IoT) that have facilitated the continuous improvement of applications such as smart health, transportation and environmental management [2]. "Smart cities" is a multidimensional term, and there is not a consensus on its definition [3]. However, there are prerequisites for smart cities, such as social, environmental and economic sustainable development, and improvement of society's living standards by using disruptive technologies [4]. Given the short cycles of technological development, it is relevant to know where these developments are heading with respect to smart cities. Notable researchers have taken some steps ahead, claiming that while the literature is ripe with descriptions of pioneering cities, there is far less systematic research into why some cities are more advanced than others [5]. In recent years, a large number of systematic studies on smart cities have been performed, bringing together contributions that can provide guidelines for further research in the field of smart cities. A study similar to this article was recently published by Esashika et al. [6], who provided a systematic review and meta-synthesis of smart cities. The authors found interesting results, such as a convergence in the literature on the primary characteristics of smart cities. The research offered a systematic and robust understanding of smart cities. However, a meta-review supported by reporting protocols (e.g., critical appraisal program) was not presented by these authors. Therefore, this article

---

intends to fill this gap in the literature, by carrying out a meta-analysis supported mainly by the CASP (Critical Assessment Skills Program), which is a critical assessment tool for systematic reviews. Meta-reviews are known to summarize evidence on a subject [7], depending on the information provided by existing reviews and their quality [8]. Thus, this meta-review was able to gather, synthesize and evaluate the existing literature, while the results are expected to influence future research, practice and policy [7]. That said, this research answers the following research question: Where are smart cities heading? Despite the variation in the research purpose and scope, a consistent message from the meta-review is that the continuous growth of digital technologies has contributed to the development of innovative product-services that seek solutions to environmental, political, economic and social challenges. In line with Esashika et al. [6], we identify topics that are relevant to smart cities, namely the dimensions, digital transformation, sustainability and resilience of smart cities. Preliminary results also suggest that research on smart cities has been monodisciplinary and very focused on megacities in developed nations. Therefore, future research should be more comprehensive and address smaller cities in developing countries.

The next section of this article focuses on the description of the methods employed, highlighting the PRISMA protocol (Preferred Reporting Items for Systematic Review and Meta-Analysis Protocol) and the content analysis technique. Section 3 presents the results of the analysis, including the state of the art, supported by a report (see Table A1, Appendix A) and a Critical Assessment Skills Program (see Table A2, Appendix A). A discussion of the results follows, supported by VOSviewer 1.6.18 (https://www.vosviewer.com, accessed on 10 July 2022), aiming to provide guidelines for future research. The last section presents the conclusions, summarizing the theoretical, managerial and political contributions.

## 2. Materials and Methods

### 2.1. Search Strategy

The search was carried out in Elsevier Scopus, with article title, abstract and keywords to identify peer-reviewed systematic reviews in English, including reviews in press, and ahead of print. Search terms were "smart cit*" AND "review*" OR "meta-analy*". In the Scopus search toolbar, we used an asterisk ("smart cit*" and "meta-analy*") to include spelling variations. For example, the asterisk allows to include "smart city" or "smart cities" as a search term. In June 2022, Scopus identified 2865 documents. The coverage ranged from 2000 to mid-2022, with an exponential growth identified in 2014. The identified documents were mainly conference proceedings (40%) and articles (30%) from India, the USA and China, in the areas of computer science (26%), engineering (20%) and social sciences (12%).

### 2.2. PRISMA Protocol

Given the high number of hits, we restricted the Scopus search to document titles. Moreover, to further refine the search, a PRISMA protocol was used. Generally, meta-reviews employ more than one database. However, using a single database, it is possible to more objectively achieve the characteristics that differentiate a systematic review, such as being transparent, replicable and easily accessible [9]. Scopus was selected since it is considered the largest international and multidisciplinary research database of peer-reviewed manuscripts [10]. The Scopus option has also been taken by other researchers publishing articles [10,11] or conference papers [12,13] on smart cities. Another argument that justifies the use of Scopus is the coverage of journals in the area of Natural Sciences and Engineering [14], areas typically associated with smart cities.

Figure 1 shows that 286 manuscripts were identified. Reviews not written in English or not published in peer-reviewed journals were excluded. This meta-review also covers the literature from 2020 to mid-2022, as most of the quality articles were published in this period. Once this process was completed, 54 reviews were obtained. In the eligibility phase, all manuscripts were carefully read in order to exclude articles for which we did not have access to full text ($n = 4$) and those that did not include systematic review and

meta-analysis as a research strategy ($n = 21$). Some exceptional cases were considered. That is, even though the words "systematic review" were not identified, articles that followed a systematic procedure (e.g., PRISMA) were considered for analysis. Considering that we did not include any additional articles that were not obtained in the Scopus search, we ended the search with 29 reviews. As this meta-review provides a considerable summary of the literature after 2020, relevant studies may have been left out. Therefore, this meta-review should be seen as a snapshot of a period of time. An additional limitation of this research is related to the purely theoretical nature of the results, requiring empirical validation.

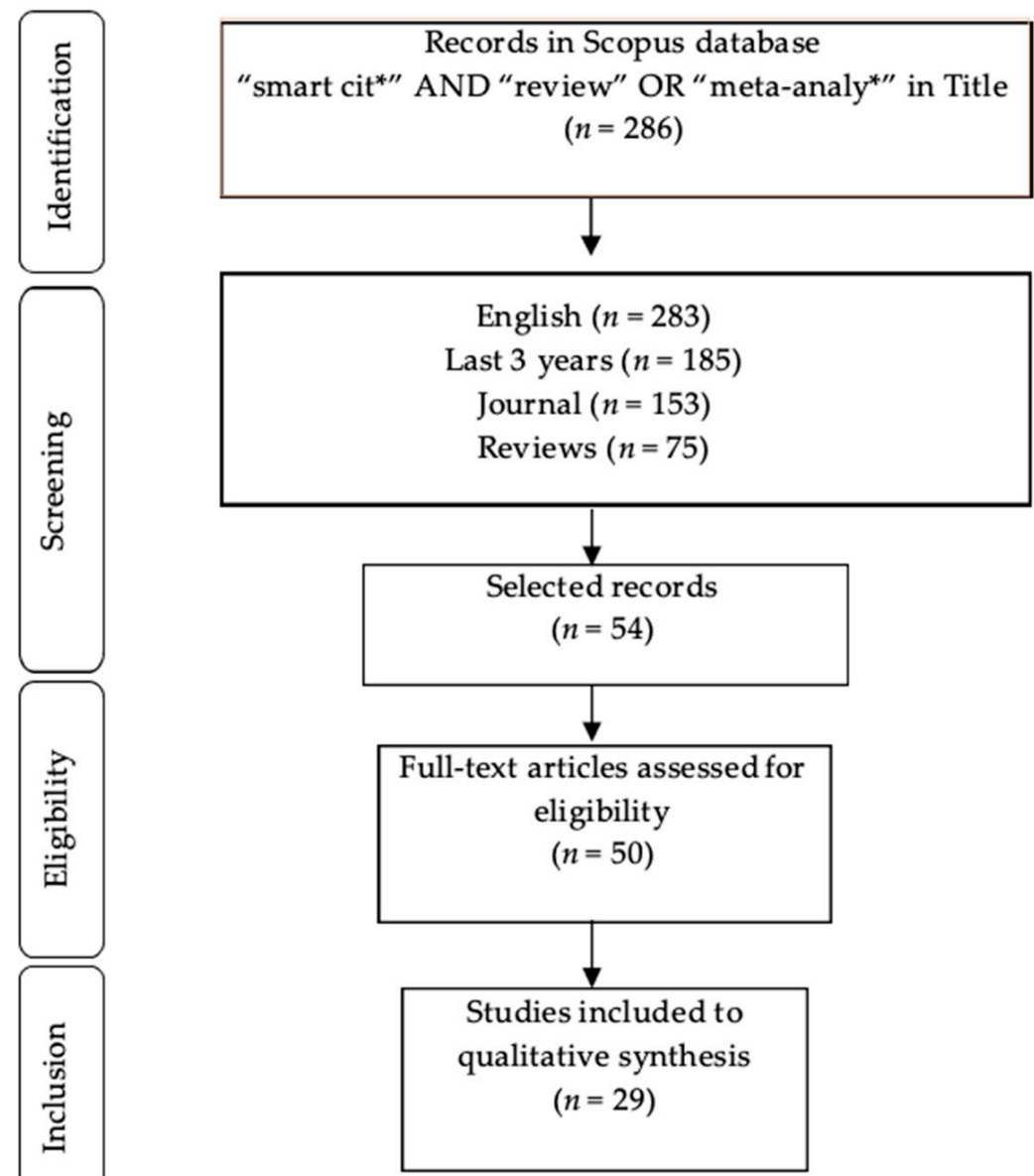

**Figure 1.** PRISMA protocol.

*2.3. Data Extraction and Synthesis*

While data were extracted from Scopus and refined by PRISMA, a report was created with the following data (Table A1): author(s), year, search period(s), search database(s), review method, and main findings. The report made it possible to synthesize the existing knowledge over the last two and half years, providing a holistic view of the subject. Moreover, a content analysis was also carried out. This technique is well known in the field of qualitative research [15] and is useful for analyzing reviews in greater detail. The process started with reading the selected reviews and adding them to NVIVO 12 (https://

www.qsrinternational.com, accessed on 24 June 2022). Analyzing qualitative data involves reading a large amount of text, looking for similarities or differences and then later finding themes and developing categories [16]. Thus, the software made it possible to integrate, code and analyze a large volume of data, dividing it into (sub)categories [17]. In more detail, the process began with identifying ideas and coding relevant words and phrases. Once this process was complete, the categories and subcategories were identified. These made it possible to identify emerging patters and ideas in the codes, in order to generate a map that provided a more detailed view of the data. The use of NVIVO 12 as a qualitative data analysis program was valuable for a more efficient analysis, being suggested by researchers who argue that this software strengthens qualitative research [18,19]. The information resulting from the content analysis is presented in Section 3 of this article. This process was also supported by VOSviewer 1.6.18, which allowed the construction and visualization of co-occurrence networks of keywords extracted from selected articles.

## 2.4. Quality Assessment

The methodological quality was assessed using the Critical Appraisal Skills Program (CASP), which is a qualitative checklist of scores for the included reviews. To do so, we used the protocol available on the CASP website (https://casp-uk.net, accessed on 3 July 2022), which already provides information regarding systematic reviews [20]. CASP analyzes the selected reviews using 10 possible items (see the footnote of the Table A2). For reliability reasons, 20% of the review studies were randomly selected for independent classification by an external researcher. CASP was very useful in selecting the high-scoring reviews and identifying the most relevant themes for the analysis. Finally, the exercise of critical evaluation in qualitative studies is of paramount importance in terms of validity [21], hence our choice to use the CASP.

## 2.5. Summary of the Methodological Process

Figure 2 graphically presents the methodological process for the meta-analysis. The materials and methods used in the selected articles (*n* = 29) are very relevant, as they accurately present the degree of maturity in smart cities. Most of the articles used PRISMA and, in some (few) cases, bibliometric analysis, as shown in Table A1. To the best of our knowledge, this is the first meta-review. Therefore, the fundamental concept behind the design of the methodological process in this article was inspired by notable researchers who have followed a similar method. For instance, Cheng and Zhang [10] performed a comprehensive meta-review of systematic reviews and meta-analyses. To this end, they carried out a systematic literature review supported by AMSTAR II, which is a critical appraisal tool for systematic reviews that include randomized or non-randomized studies of healthcare interventions, or both [22]. Another similar study was presented by O'Connor et al. [23], who performed a meta-analysis supported by PRISMA and assessed by CASP. Although we highlight the authors identified above, many other meta-reviews have followed a combination of protocols and content analysis/software [24–26].

The methods used are linked to the articles we reviewed, insofar as: (1) they allowed us to select the most suitable manuscripts from a vast literature; (2) they clearly define topics related to smart cities; (3) they develop a conceptual model; and (4) they define a research agenda (Figure 2). The next section will present the results of the article and explain in more detail each of the previous four points.

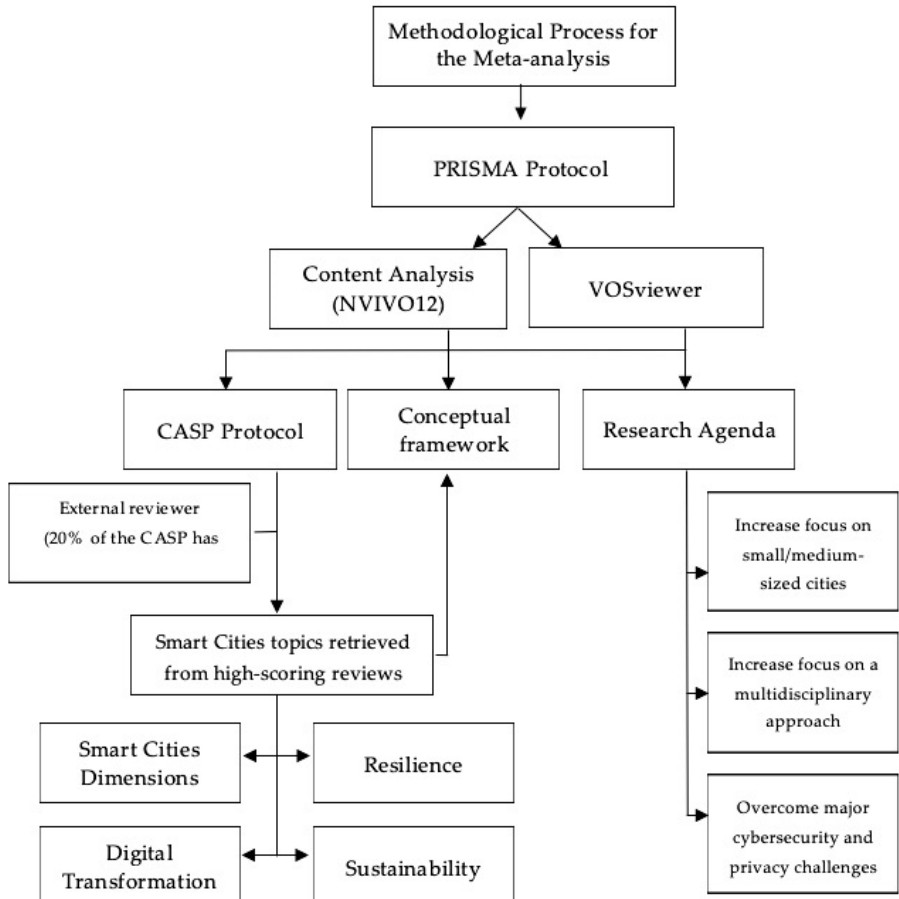

**Figure 2.** Methodological process.

### 3. Results

The publisher of peer-reviewed journals with the highest percentage (55%) of article publication was the Multidisciplinary Digital Publishing Instituto (MDPI). In addition, 62% of the published articles were Scopus Q1. However, the average CASP score (Table A2) is 16 pts, which means the overall quality is good (on a 4-level scale: excellent, good, moderate, poor). Only 24% of the articles scored excellent [27–33], 60% good and 17% of moderate quality. Most of the generally high-scoring reviews (i.e., ≥16%) published in Q1 journals focused primarily on topics such as smart cities dimensions [27,29], digital transformation [31,32], sustainability [30,32,34–37] and resilience [30,38,39]. With lower scores (i.e., <16%), we found technical themes such as IoT [40] and sensors [35] for sustainable smart cities. The same does not apply to augmented reality [41], artificial intelligence [42] and cyber [39,41], which achieved higher scores, even when published in Q2 journals. This phenomenon is partially explained as the authors may have had difficulty finding relevant studies on these specific technical themes, resulting in less relevant findings when compared to others. In particular, as we will see further on, research in smart cities is monodisciplinary (i.e., too focused on certain research areas). Based on the above information, we present the conceptual framework of smart cities (Figure 3). The conceptual framework was basically developed using the information from articles that achieved a high CASP score and were published in Scopus Q1 journals. The description of each topic is presented in the following subsections.

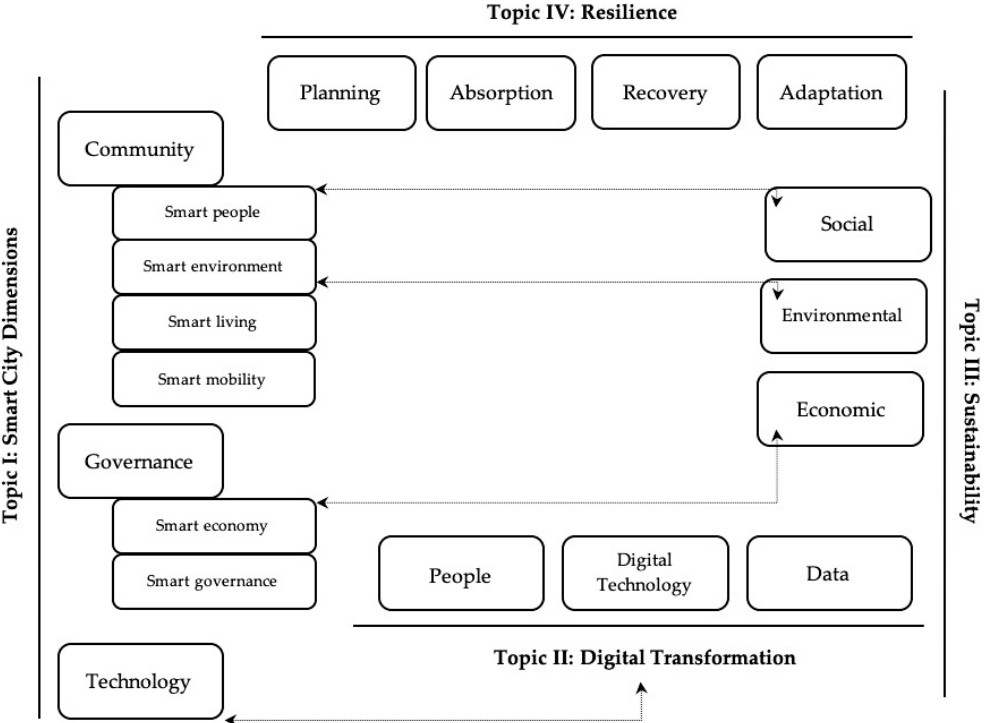

**Figure 3.** Conceptual framework.

### 3.1. Smart Cities Dimensions

There is no consensus in the literature on the dimensions of smart cities. However, there is some degree of agreement in the selected dimensions (Figure 3). Sharif and Pokharel [27] identified six dimensions, namely "smart economy", "smart governance", "smart living", "smart mobility", "smart people" and "smart environment", while Kasznar et al. [29] identified three main dimensions, "technology", "community" and "governance". Grouping the dimensions identified by Sharif and Pokharel [27] and by Kasznar et al. [29], we place special emphasis on (1) "community", which includes smart life, smart mobility and smart people, and (2) "governance" where we focus on smart governance and smart environment. As far as technology is concerned, we frame it as being directly related to digital transformation, which is one of the macro topics analyzed in the article.

This section follows a top-down approach to the conceptual framework, starting with a discussion of the "community" dimension (Figure 3).

In the literature, it is evident that "social infrastructures" (topic III) of smart cities have a strong relationship with human and social capital. Human capital (i.e., skills and proficiencies of a person or group) and social capital (i.e., number and quality of relationships that connect social organizations) are critical, as they foster innovation. In this regard, universities stood out in the literature, as they are organizations formed by highly qualified people (i.e., "intelligent people"). Intelligent people usually stimulate human capital, and tend to develop disruptive technologies, transferring them to public and private organizations. E-participation and e-government are good examples, as they are based on the latest technologies, such as AI. These technologies have improved the participation of citizens in general, as they make cities more efficient in terms of decision making by policymakers and government service delivery. The "smart environment", in addition to being part of the community dimension (topic I), is also directly related to the "sustainability" of smart cities (topic III). This dimension includes a series of waste management strategies, pollution control, etc. That is, the use of technology assists in the preservation of natural resources through sustainable methods and real-time data collection. These methods are helping decision makers to optimize the classification, collection, recycling and reuse of waste. The Organization for Economic Co-operation and

Development (OECD) considers the development and preservation of natural, economic and human capital to be essential elements for "smart living". All factors that make contemporary human life possible, such as social well-being and associated technologies, can be included in this sub-dimension. "Smart living" is also broad, as it spans from using smart home applications (e.g., home automation) to the workplace (e.g., grid-connected and internet-enabled lighting and security). These applications may also collect personal and private data about their users, incurring privacy and security risks. Lastly, "smart mobility" is often focused on transport systems and infrastructure, as frequent problems in cities are related to congestion, queues and delays. In this sub-dimension, the widespread use of connectivity (e.g., IoT) has been key. It provides real-time data to determine the best routes, allowing traffic efficiency and greater security.

Regarding "governance", it can be subdivided into two relevant issues (Figure 3).

The "smart economy" comprises the guidelines and policies that inspire innovation and creativity. Innovation is associated with scientific research and disruptive technologies, with special attention given to sustainability (topic III). In general, the smart economy makes use of information and communication technologies (ICT) to improve the economic element of a community and the socially responsible use of resources. The smart economy has unique characteristics, challenges, and solutions, requiring disciplined development in different areas such as science, industry and business. The second issue is "smart governance", which, through political and strategic decisions, is associated with decision-making for a better provision of public and social services. Governance is therefore seen as the coordination between the citizens and administrative institutions of a state. To maximize efficiency, smart cities integrate private and public servers, ensuring that all city services and resources are served through high-tech solutions. Electronic governance has been one of the main pillars in the context of collective efforts that allow the development of effective interactions between all actors in smart cities.

As the "technology" dimension has been identified by many authors as part of digital transformation (Figure 3) [31,43,44], it will be addressed in the following section.

*3.2. Digital Transformation*

Anthony [31] distinguished the stages of digital transformation for smart cities (i.e., digitization, digitalization and digital transformation). The aforementioned stages are not very different from the general literature on digital transformation [43–46]. What is new is that the recent growth of digital technologies is allowing cities to undergo a transformation that allows them to optimize smart services and offer new products. In this regard, there have been numerous special issues [47], encouraging these initiatives in smart cities. Therefore, we generally found that the continuous growth of digital technologies has contributed to the development of innovative products-services that seek solutions to environmental, political, economic and social challenges. The great advantage of digital transformation continues to be disruption, that is, the interruption of traditional business models in different sectors, aiming to realign processes, technologies, and business models in order to create value for customers and companies. However, academics are arguing that products-services are not integrated in smart cities as desired, such that they can provide a value-added network. Smart cities remain under increasing pressure to thrive in ever-changing environments. Dynamics in both the economy and technology pose serious challenges for cities, as there is a need to adapt to complex changing conditions and ensure system integration. However, while digital transformation allows humans to cooperate with autonomous systems, the structural challenges are high, especially in cities that are composed of different entities with different technological and social structures. Although the literature examines the digital transformation in the domain of smart cities, it still does not investigate how the complexity and integration of the system can be improved [31].

*3.3. Sustainability*

In the literature on smart cities, sustainability is discussed by Lopez and Castro [30], Zheng et al. [32], and Cortese et al. [34].

Lopez and Castro [30] stress that cities are complex systems that work like a gear, where the relationship between interurban and intraurban processes is the key factor in understanding their synchronization. In turn, smart cities should promote the integration and interconnection of systems to offer better services and increase citizens' quality of life. The transformation process must be observed from the perspective of urban ecology, converging on the concept of urban ecosystems (symbiosis between the natural and the spatial). Sustainability should not revolve strictly around a principle of economic growth and territorial expansion, but rather an approach to sustainable development that seeks to balance ecosystems. For Lopez and Castro [30], it is clear that sustainability requires results through the generation of tools that drive a long-term transformation of society, while sustainable development is the roadmap to be established, allowing for the adaptation and proactive reorganization of institutions and public policies. Achieving sustainable urban development depends on the quality of the environment that provides ecosystem services. However, the symbiosis of technology and planning offers a certain path to innovation in smart environmental planning [48]. Within the triad of social, environmental and economic aspects, Lopez and Castro [30] focus their discussion on the environmental and social pillars.

Zheng et al. [32] argue that there appears to be a lack of systematic quantitative and visual investigation and multidisciplinary scrutiny of the structure and evolution of smart cities. They also point out that ICTs have the potential to contribute to sustainability within the discourse of the smart sustainable city. Focusing on sociocultural and political–institutional structures inserted in the development of smart cities. In summary, sustainable smart cities present a new avenue of theoretical and practical research with which to explore the potential of ICT applications to contribute to urban sustainability.

In addition to the traditional dimensions (i.e., social, environmental, economic), Cortese et al. [34] highlight the sustainable understanding of energy in the context of smart cities. They also mention that while there is a strong concentration on the technological dimension of sustainability, energy efficiency, and renewable energy topics in the literature, much less attention is paid to urban planning issues. This is in line with other authors with regard to the strong focus on the technological and smaller dimension in urban planning.

In general, the authors slightly depart from the economic and technological issues of sustainability. In turn, they emphasize the balance of ecosystems and the long-term transformation of society. Focusing on urban planning, that is, on the integration of inter-urban and intra-urban processes, in order to better interconnect systems and technologies in order to offer better products and services. Thus, the future is expected to be in the environmental and social issues of sustainability, in order to promote an increase in the quality of life of citizens [49].

*3.4. Resilience*

With regard to resilience in smart cities, we identified three approaches, namely those of Sharifi et al. [38], Lopez and Castro [30], and Ahmadi-Assalemi et al. [39]. Although the approach by Sharifi et al. (2021) is, in our view, the most complete and is the one adopted in this article, there are significant points of agreement between the different authors.

The term "resilience" has been more associated with smart cities since the emergence of COVID-19. According to Sharifi et al. [38], Smart Cities' solutions and technologies are focused on four stages: (1) planning and preparedness (pre-disaster); (2) absorption (during the disaster); (3) recovery (post-disaster); and (4) adaptation (review). Examples include the fact that the (1) investment in planning and adoption of smart solutions and technologies has increased the ability of cities to predict patterns in the transmission of the SARS-CoV-2, minimizing its spread; (2) absorption was related to the use of smartphone apps to detect

individuals with symptoms, such as measures taken to send alerts to those who do not follow emergency protocols and those who may be spreading the virus, reducing the speed of its transmission; (3) recovery is related to measures that allowed communities to return to the pre-shock state, contributing to the recovery process; and, finally (4) that its adaptation refers to the ability to take advantage of the adverse event as an opportunity to improve short- and long-term overall performance.

Lopez and Castro [30], in turn, divide resilience into three stages that are very similar to those mentioned above, namely: (1) change; (2) adaptation; and (3) transformation. These authors consider that the adaptive cycle does not occur in a short-term period, and the ability to respond occurs in the long term. Resilience analysis lacks assessment tools for decision makers in government agencies that make it possible to establish the ability to adapt to a complex system that evolves over time. Furthermore, increasing a city's resilience makes it more sustainable, but increasing a city's sustainability does not necessarily make it more resilient.

Ahmadi-Assalemi et al. [39] analyzed the more technical components of resilience, such as cyber resilience. This issue is related to the rapid growth of smart cities, including the use of emerging and innovative technologies that create highly fragile and complex cyber-physical-natural ecosystems.

Overall, the most appropriate approach to resilience in smart cities seems to be more appropriately divisible into four phases, as presented by Sharifi et al. [38]: (1) planning and preparedness (pre-disaster); (2) absorption (during the disaster); (3) recovery (post-disaster); and (4) adaptation (review).

## 4. Discussion

This section presents a VOSviewer 1.6.18 print that shows the importance of developing a research agenda. Future research may also include the use of other tools, such as using the Gephi network. The VOSviewer 1.6.18 data was retrieved using Zotero, which produced an .RIS file with the results presented below. Figure 4 shows the clusters of selected articles and the need to identify the "research trends", "initial models", and "architectures" of smart cities through systematic reviews (PRISMA) and/or bibliometrics. Thus, this meta-analysis makes it possible to aggregate dispersed and heterogeneous information, making it available to the reader/researcher in an organized and clean way.

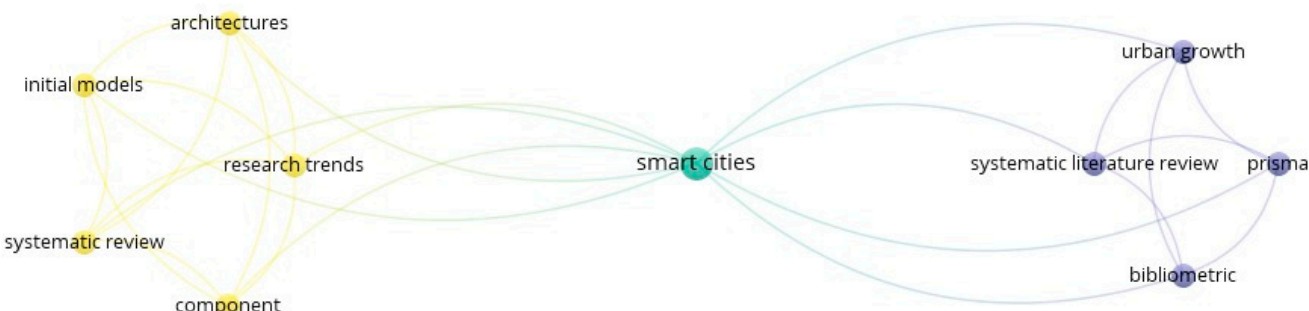

**Figure 4.** VOSviewer 1.6.18 analysis.

We focus our attention on defining the research trends for smart cities. To do so, we retrieved future suggestions from the 29 selected articles. Some of the proposals for further research are given below.

### 4.1. Geographical Configuration

Pratama [50] stressed that the geographic configuration of smart cities is polarized, with greater attention being given to megacities in developed nations. Therefore, one of the recommendations is to focus the scientific research on smart cities in small- and medium-sized cities in developing countries, with the aim of balancing academic discourse and debates. As per the recommendation of Christofi et al. [51], the need for comparative

studies in developing economies was stressed. As an example, research can be carried out by comparing Asia and smart cities in the European Union or North America. Comparing different regions and cities will make it possible to achieve a comprehensive understanding of smart city strategies, providing vital insights aimed at increasing generalizability and greater scalability of results.

### 4.2. Multidisciplinary Approach

It should be noted that the natural sciences have dominated the production and impact of research compared to the social sciences [50]. That is, the available literature shows that empirical studies of smart cities have been dominated by a monodisciplinary perspective. This may limit the search for smart cities, with fewer sociocultural artifacts and humanity attributes. In this regard, Rozario et al. [10] suggest focusing future studies on a multidisci­plinary approach, reorienting the three pillars (people, processes and technology) necessary for the successful implementation of the smart city concept. Meanwhile, Pratama [50] dis­tances themselves from technological determinism and techno-singularity. Nevertheless, there is a consensus that insights from the humanities and the social sciences can enrich the smart city research landscape. Alternative interdisciplinary or transdisciplinary approaches are likely to offer fruitful insights.

### 4.3. Cybersecurity and Privacy Challenges

We found that recent empirical research, combined with a strong theoretical framework produced by academics and practitioners, suggests that it is crucial to overcome major cybersecurity and privacy challenges in smart cities [27]. The continuous interconnectivity in smart cities can give rise to a series of cybersecurity threats, such as the loss of privacy and confidentiality [38,43], "physical threats, systems and applications vulnerability, malware injection attacks, denial of service (DoS), malicious insider threats, and data leakage" [41]. Thus, it is suggested that future research should highlight the role played by cybersecurity in smart city development and analyze the impact of smart city development on the link between city and stakeholders, and the dynamics of inner city development [51,52]. Lastly, it should be noted that this study highlights research directions for smart cities; however, more robust empirical evidence is needed to validate the results presented in this article.

### 5. Conclusions

The main objective of this research was centered on the question: where are smart cities heading? Although it is quite challenging to answer such a vast question, this article brings together several theoretical, political, and practical contributions.

One of the original and innovative theoretical contribution of this research is the development and discussion of a conceptual framework for smart cities (Figure 3). To this end, we used a meta-analysis, a series of protocols including PRISMA and CASP, and software such as VOSviewer 1.6.18 and NVIVO 12. These tools made it possible to identify the most relevant topics for smart cities, i.e., (1) smart cities dimensions; (2) digital transformation; (3) sustainability; and (4) resilience.

Regarding the first topic, we observed that although there is no full consensus for the dimensions, there is some level of agreement, whereby we identified "community", "governance" and "technologies" as the most relevant. When it comes to the second topic, "digital transformation", the continuous growth of digital technologies has contributed to disruption of traditional business models in different sectors, enabling the development of new products and services. However, while the realignment of processes and the creation of new models aims to create added value for customers, all the evidence leads us to believe that the product-services are not integrated in the networked smart cities or at least as desired. This issue can even be used by policy makers to stimulate public investment to create strategies for developing network structures. In this regard, the sustainability of smart cities (third topic) should not revolve strictly around a principle of economic growth and territorial expansion, but rather a sustainable development approach that seeks

the balance of ecosystems and the quality of life for citizens. Therefore, as a suggestion for policy makers and practitioners, it goes in the direction of departing some way from strictly economic and technological issues, emphasizing the balance of ecosystems and the long-term transformation of society. In fact, one of the suggestions for future research (vide research agenda) is to invest in the approach to the social sciences domain. That is, the recommendation is not distanced from the natural sciences, but rather an interdisciplinary or transdisciplinary approach, in which the social sciences are included. Lastly, regarding the fourth topic, the "resilience" of smart cities, we found that it was divided into four distinct phases: (1) planning and preparation (pre-disaster); (2) absorption (during the disaster); (3) recovery (post-disaster); and (4) adaptation (review). In this regard, the literature offers examples in the context of COVID-19 and cybersecurity.

With regard to the topics analyzed, we found that the discussion in the literature was for very specific contexts; see the case of resilience which was focused COVID-19 and cybersecurity. However, research dominated by the monodisciplinary perspective paves the way for academics to invest in multidisciplinary research. The scope for other research domains could be a way to generalize the results of the research in smart cities. For practitioners, this opportunity may be related to the empirical validation of the theoretical results, since some of the lessons learned can be put into practice in other sectors of activity.

Finally, another issue that stands out from our analysis is related to the polarization of research in smart cities. In other words, attention has been focused on the megacities of developed nations. Therefore, greater efforts should be made to carry out scientific research in small and/or medium-sized cities in developing countries.

**Author Contributions:** Conceptualization, J.R.; methodology, J.R.; software, J.R.; validation, J.R., P.A.M. and P.C.M.; formal analysis, J.R.; investigation, J.R.; resources, J.R.; data curation, J.R.; writing—original draft preparation, J.R.; writing—review and editing, J.R., P.A.M. and P.C.M.; visualization, J.R.; supervision, J.R.; project administration, J.R.; funding acquisition, J.R. All authors have read and agreed to the published version of the manuscript.

**Funding:** This research received no external funding.

**Institutional Review Board Statement:** Not applicable.

**Informed Consent Statement:** Not applicable.

**Data Availability Statement:** Not applicable.

**Conflicts of Interest:** The author declare no conflict of interest.

## Appendix A

**Table A1.** Report—Characteristics of review studies (*n* = 29).

| Authors | Year | Search Period(s) | Search Database(s) | Review Method | Article Main Findings (Excerpts Taken Directly from the Articles) |
|---|---|---|---|---|---|
| Dashkevych and Portnov [53] | 2022 | 10.2020 | Web of Science Core, Collections, Scopus, ScienceDirect | Systematic Literature Review (PRISMA) | "Identification of 48 smart city identification metrics, which are further split into three main categories-smart digital technology, living conditions and environmental sustainability." |
| Alzahrani and Alfouzan [41] | 2022 | 9.2021 | Emerald Insight, Science Direct, IEEE Xplore | Systematic Literature Review (PRISMA) | "The study identified five main categories of AR and cybersecurity applications for smart cities, which can be classified within tourism, monitoring, system management, education and mobility." |
| Cortese et al. [34] | 2022 | Not mentioned | Web of Science, Scopus | Systematic Literature Review (PRISMA) | "The research identified a global publication landscape for smart city and energy sustainability research; unbalanced publications when critically evaluating geographical continent's energy use intensity vs smart cities energy; heavy concentration on the technology dimension of energy sustainability and efficiency and renewable topics in the literature but less attention to the energy and urban planning." |
| Rocha et al. [28] | 2022 | 7.2021 | Scopus, Web of Science, IEEE Xplorer | Systematic Literature Review (PRISMA) | "The results show the interest in using context-aware features to develop smart cities' applications targeting public health, tourism experience, urban mobility, active citizenship, shopping experience, management of urban infrastructures, public alerts, recommenders, and smart environments." |
| Bellini et al. [40] | 2022 | 10.2021 | Web of Science | Systematic Literature Review | "From this article it emerged that in recent years, the integration of IoT solutions and smart city frameworks is achieving increasingly higher levels of complexity and wider application ranges, which go beyond the past generation of vertical silo applications that were based on specific domains." |

Table A1. *Cont.*

| Authors | Year | Search Period(s) | Search Database(s) | Review Method | Article Main Findings (Excerpts Taken Directly from the Articles) |
|---|---|---|---|---|---|
| Sharif and Pokharel [27] | 2022 | Not mentioned | ScienceDirect, Scopus, IEEEXplore, Taylor & Francis, Wiley | Systematic Literature Review | "The findings of the literature review illustrate that not all smart cities adapt all of the smart city dimensions. The dominant technology used in smart cities' applications is found to be the Internet of Things, Artificial Intelligence, and blockchain." |
| Arief et al. [54] | 2022 | Not mentioned | Scopus, IEEE Xplore, ACM databases, Science Direct, Springer | Systematic Literature Review Questionnaire | "The systematic review results show popular topics, such as the standardization of a smart city and the strategies used to determine relevant models in each city's uniqueness and context. The results also identified thirteen smart city components and their challenges. Furthermore, this study's novelty is proposed the smart city's initial models with the smart government as a key component and to be a centre of other smart city components." |
| Hurbean et al. [55] | 2021 | 6.2021 | Web of Science, Scopus, IEEE Xplore, AIS, Springer, Proquest, MDPI, Semantic Scholar | Systematic Literature Review (PRISMA) | "The results revealed that: (a) machine learning applications using open data came out in all the SC areas and specific ML techniques are discovered for each area, with deep learning and supervised learning being the first choices. (b) Open data platforms represent the most frequently used source of data. (c) The challenges associated with open data utilization vary from quality of data, to frequency of data collection, to consistency of data, and data format." |
| Rozario et al. [10] | 2021 | Not mentioned | Scopus | Systematic Literature Review | "This research found that the fields of specialisations such as information technology and infrastructure engineering in contributing to smart cities need a cross-domain holistic approach of managing people-centric service requirements for improving consumer satisfaction and sustainability." |
| Ramírez-Moreno et al. [35] | 2021 | Not mentioned | Scopus, Google Scholar, IEEE Xplore | Systematic Literature Review (PRISMA) | "This article found that although the use of these sensors is diverse, their application can be categorized in six different groups: energy, health, mobility, security, water, and waste management." |

**Table A1.** *Cont.*

| Authors | Year | Search Period(s) | Search Database(s) | Review Method | Article Main Findings (Excerpts Taken Directly from the Articles) |
|---|---|---|---|---|---|
| Mills et al. [36] | 2021 | Not mentioned | WoS (Core Collection), ProQuest Central, EBSCOhost | Systematic Literature Review (PRISMA) | "The article stresses that in addition to smart theory of the SCCF framework and the attributes of collaboration will assist theorists and practitioners to build more effective smart city prescriptions and practice." |
| Rocha et al. [56] | 2021 | 4.2021 | Web of Science, Scopus, IEEE Xplore | Systematic Literature Review (PRISMA) | "One of the main findings is that the number of included articles is reduced when compared with the total number of articles related to smart cities, which means that the mobility of older adults it is still a not significant topic within the research on smart cities'." |
| Sharifi et al. [38] | 2021 | 10.2020 | Scopus | Systematic Literature Review (PRISMA) | "The review shows that investment in smart city initiatives can enhance the planning and preparation ability. In addition, the adoption of smart solutions and technologies can, among other things, enhance the capacity of cities to predict pandemic patterns, facilitate an integrated and timely response, minimize or postpone transmission of COVID-19, provide support to overstretched sectors, minimize supply chain disruption, ensure continuity of basic services, and offer solutions for optimizing city operations." |
| Pratama [50] | 2021 | Not mentioned | Web of Science, Google Scholar | Systematic Literature Review (PRISMA) | "This article stresses that there is a fragmentation of smart city research and lack of intellectual discussion among disciplines. Thus, the available literature shows that smart city empirical studies were dominated by mono-disciplinary perspective. Thus, conventional academic tradition is in the forefront in researching smart cities in real worlds." |
| Nicola and Villani [57] | 2021 | 1.2021 | Scopus | Systematic Literature Review | "This article proposed a classification of the sub-domains of the city addressed by the ontologies we found, and the research issues that have been considered so far by the scientific community." |

**Table A1.** *Cont.*

| Authors | Year | Search Period(s) | Search Database(s) | Review Method | Article Main Findings (Excerpts Taken Directly from the Articles) |
|---|---|---|---|---|---|
| Kim et al. [37] | 2021 | Not mentioned | Scopus, IEEE Xplore | Systematic Literature Review | "This study suggests that the following innovative solutions be suitably applied to advanced energy conservation systems in sustainable smart cities: (i) construction of infrastructure for advanced energy conservation systems, and (ii) adoption of a new strategy for energy trading in distributed energy systems." |
| Kasznar et al. [29] | 2021 | Not mentioned | Scopus, EmeraldInsight, IEEE Xplore | Systematic Literature Review, Bibliometric and Blibliographic Analysis | "The bibliographic analysis reflected major aspects of smart city infrastructure, including: IT infrastructure, sustainable and ecological buildings, urban systems, smart initiatives, and applications that stimulate e governance and e-participation." |
| Wang et al. [58] | 2021 | 11.2018 | Web of Science Core, Collection database | Systematic Literature Review | "This article illustrates the relationship among data, research, and policy application, identifying the roles of researchers in computer science and geography, practitioner in market or government and policy makers in promoting smart application." |
| Christofi et al. [51] | 2021 | Not mentioned | EBSCO | Systematic Literature Review | "Building on the antecedents–phenomenon–consequences framework, the authors discuss the antecedents and consequences of the various innovative marketing strategies that smart cities adopt for their internationalization and development of an international competitive advantage. In the process of doing so, the authors synthesize the findings of the studies as well as literature gaps that provide fruitful avenues for future research." |
| Lopez and Castro [30] | 2021 | Not mentioned | Scopus, Web of Science, Google Scholar | Bibliometric Research Systematic Literature Review (PRISMA) | "The main result is to consider cities with a complex systems approach that works like a gear; the relationship between inter-urban and intra-urban processes is the key factor that allows for an understanding of their synchronization." |

| Authors | Year | Search Period(s) | Search Database(s) | Review Method | Article Main Findings (Excerpts Taken Directly from the Articles) |
|---|---|---|---|---|---|
| Anthony [31] | 2021 | 10.2019 | Scopus, Web of Science | Bibliometric Research Systematic Literature Review (PRISMA) | "Municipalities still struggle with managing data integration and complexity. Accordingly, this study systematically reviews 70 research articles from 1999 to 2020 and discusses on development and state-of-the-art of Enterprise Architecture (EA) and digital transformation of cities into smart cities." |
| Ahmadi-Assalemi et al. [39] | 2020 | 4.2019 | IEEE, ACM DL, Science Direct, WoS and Scopus | Systematic Literature Review | "The article shows that CPSs addressing cyber resilience and support for modern DFIR are a recent paradigm. Most of the primary studies are focused on a subset of the incident response process, the "detection and analysis" phase whilst attempts to address other parts of the DFIR process remain limited. Further analysis shows that research focused on smart healthcare and smart citizen were addressed only by a small number of primary studies." |
| Zheng et al. [32] | 2020 | 3.2019 | WoS | Scientometric Review | "The article used a scientometric technique that: (1) reveal the intellectual division of this developing field using a visual and comprehensive approach, (2) identify in chronological order the 10 core research sub-topics in this area with burst references and terms, (3) identify Internet of Things, big data, and fog computing as the most promising technologies for SC planning and development, and (4) conclude that smart sustainable cities and sustainable smart cities are the two emerging trends in the domain." |
| Prasetyo and Lubis [59] | 2020 | Not mentioned | Not mentioned | Systematic Literature Review, Meta-analysis | "The research discussed the Enterprise Architecture (EA) research overview on smart city design and the gaps in EA implementation for smart city architecture development." |

| Authors | Year | Search Period(s) | Search Database(s) | Review Method | Article Main Findings (Excerpts Taken Directly from the Articles) |
|---|---|---|---|---|---|
| Iskandaryan et al. [60] | 2020 | Not mentioned | Scopus, IEEE Xplore | Systematic Literature Review | "As a result, the paper concludes that: (1) instead of using simple machine learning techniques, currently, the authors apply advanced and sophisticated techniques, (2) China was the leading country in terms of a case study, (3) Particulate matter with diameter equal to 2.5 μm was the main prediction target, (4) in 41% of the publications the authors carried out the prediction for the next day, (5) 66% of the studies used data had an hourly rate, (6) 49% of the papers used open data and since 2016 it had a tendency to increase, and (7) for efficient air quality prediction it is important to consider the external factors such as weather conditions, spatial characteristics, and temporal features." |
| Buttazzoni et al. [33] | 2020 | 5.2019 | CINAHL, PsycINFO, PubMed database, Elsevier's Scopus and Web of Science | Systematic Literature Review | "28 articles were retained, assessed, and coded for their inclusion of equity characteristics using the Cochrane PROGRESS-Plus tool (referring to (P) place of residence, (R) race, (O) occupation, (G) gender, (R) religion, (E) education, (S) socio-economic status (SES), and (S) social capital). The most frequently included equity considerations in smart city health interventions were place of residence, SES, social capital, and personal characteristics; conversely, occupation, gender or sex, religion, race, ethnicity, culture, language, and education characteristics were comparatively less featured in such interventions." |
| Saharan et al. [61] | 2020 | Not mentioned | ACM Digital Library, Springer, IEEE eXplore, Wiley Interscience, Taylor and Francis, and ScienceDirec | Systematic Literature Review | "An in-efficient dynamic pricing technique may lead to the mismanagement of vehicles, which results an increase in the waiting time of vehicles, an increase in air and noise pollution, wastage of electric and other sources of energies. Various problems solved by the dynamic pricing techniques, importance of various evaluation parameters, limitations of dynamic pricing techniques and their applications are discussed in-depth in the paper." |

**Table A1.** *Cont.*

| Authors | Year | Search Period(s) | Search Database(s) | Review Method | Article Main Findings (Excerpts Taken Directly from the Articles) |
|---|---|---|---|---|---|
| Ahmed et al. [62] | 2020 | Not mentioned | ACM Digital Library, Springer, IEEE eXplore and ScienceDirec | Systematic Literature Review | "The article explores and identifies the barriers and hurdles in Smart City Domain and how these hurdles are mitigated by the blockchain technology." |
| Yigitcanlar et al. [42] | 2020 | 12.2019 | Directory of Open Access Journals, Science Direct, Scopus, TRID, Web of Science, and Wiley Online Library | Systematic Literature Review | "The findings of the systematic review containing 93 articles disclose that: (a) AI in the context of smart cities is an emerging field of research and practice. (b) The central focus of the literature is on AI technologies, algorithms, and their current and prospective applications. (c) AI applications in the context of smart cities mainly concentrate on business efficiency, data analytics, education, energy, environmental sustainability, health, land use, security, transport, and urban management areas. (d) There is limited scholarly research investigating the risks of wider AI utilization. (e) Upcoming disruptions of AI in cities and societies have not been adequately examined." |

**Table A2.** Methodological quality ratings based on CASP (*n* = 29).

| Authors | Item 1 | Item 2 | Item 3 | Item 4 | Item 5 | Item 6 | Item 7 | Item 8 | Item 9 | Item 10 | Score | Classification Quality | Scimago |
|---|---|---|---|---|---|---|---|---|---|---|---|---|---|
| Dashkevych and Portnov [53] | N | Y | C | N | C | Y | Y | C | Y | Y | 14 | Moderate | Q1 |
| Alzahrani and Alfouzan [41] | Y | Y | Y | N | Y | Y | C | C | Y | Y | 16 | Good | Q1 |
| Cortese et al. [34] | Y | C | C | N | Y | Y | Y | Y | Y | Y | 16 | Good | Q1 |
| Rocha et al. [28] | Y | Y | Y | Y | C | Y | C | C | Y | C | 16 | Good | Q2 |
| Bellini et al. [40] | N | C | C | C | C | N | C | C | Y | Y | 10 | Moderate | Q2 |
| Sharif and Pokharel [27] | Y | C | Y | Y | Y | Y | Y | C | Y | Y | 18 | Excellent | Q1 |
| Arief et al. [54] | Y | Y | C | Y | C | Y | Y | Y | C | C | 16 | Good | Q4 |
| Hurbean et al. [55] | Y | C | C | Y | C | Y | Y | C | Y | Y | 16 | Good | Q2 |
| Rozario et al. [10] | Y | C | C | N | Y | Y | Y | C | Y | Y | 15 | Good | Q2 |
| Ramírez-Moreno et al. [35] | N | Y | C | Y | C | Y | C | C | Y | Y | 14 | Moderate | Q2 |
| Mills et al. [36] | Y | C | N | N | C | Y | C | C | Y | C | 11 | Moderate | Q1 |
| Rocha et al. [56] | Y | Y | Y | Y | C | Y | Y | C | Y | Y | 18 | Excellent | Q2 |
| Sharifi et al. [38] | Y | Y | Y | N | C | Y | Y | C | Y | Y | 16 | Good | Q1 |
| Pratama [50] | Y | Y | Y | Y | C | Y | C | C | C | Y | 16 | Good | Q1 |

**Table A2.** *Cont.*

| Authors | Item 1 | Item 2 | Item 3 | Item 4 | Item 5 | Item 6 | Item 7 | Item 8 | Item 9 | Item 10 | Score | Classification Quality | Scimago |
|---|---|---|---|---|---|---|---|---|---|---|---|---|---|
| Nicola and Villani [57] | Y | C | C | N | C | Y | Y | C | Y | Y | 14 | Moderate | Q1 |
| Kim et al. [37] | Y | Y | Y | N | Y | Y | C | C | Y | Y | 16 | Good | Q1 |
| Kasznar et al. [29] | Y | Y | Y | N | Y | Y | Y | Y | Y | Y | 18 | Excellent | Q1 |
| Wang et al. [58] | Y | Y | C | C | C | Y | Y | Y | Y | Y | 17 | Good | Q1 |
| Christofi et al. [51] | Y | Y | C | Y | C | Y | Y | Y | Y | C | 17 | Good | Q1 |
| Lopez and Castro [30] | Y | Y | Y | N | Y | Y | Y | Y | Y | Y | 18 | Excellent | Q1 |
| Anthony [31] | Y | Y | Y | Y | Y | Y | Y | C | Y | C | 18 | Excellent | Q1 |
| Ahmadi-Assalemi et al. [39] | Y | C | C | Y | Y | Y | Y | C | Y | Y | 17 | Good | Q1 |
| Zheng et al. [32] | N | Y | Y | Y | Y | Y | Y | Y | Y | Y | 18 | Excellent | Q1 |
| Prasetyo and Lubis [59] | Y | C | C | Y | Y | Y | Y | C | Y | Y | 17 | Good | Q2 |
| Iskandaryan et al. [60] | Y | C | C | N | Y | Y | Y | Y | Y | Y | 16 | Good | Q2 |
| Buttazzoni et al. [33] | Y | Y | Y | Y | C | Y | Y | C | Y | Y | 18 | Excellent | Q2 |
| Saharan et al. [61] | Y | C | Y | Y | C | Y | Y | C | Y | Y | 17 | Good | Q1 |
| Ahmed et al. [62] | Y | C | Y | Y | C | Y | Y | C | Y | C | 16 | Good | Q1 |
| Yigitcanlar et al. [42] | Y | C | Y | N | Y | Y | Y | C | Y | Y | 16 | Good | Q2 |
| | | | | | | | | | | Mean | 16 | | |

Abbreviations: Y = Yes; C = Can't tell; N = No | Classification: Yes = 2; C = 1 and N = 0 | Overall classification: Excellent = 18/20; Good = 15/17; Moderate 10/14; Poor ≥ 10. 1 = Did the review address a clearly focused question?; 2 = Did the authors look for the right type of papers?; 3 = Do you think all the important, relevant studies were included?; 4 = Did the review's authors do enough to assess the quality of the included studies?; 5 = If the results of the review have been combined, was it reasonable to do so?; 6 = What are the overall results of the review?; 7 = How precise are the results?; 8 = Can the results be applied to the local population?; 9 = Were all important outcomes considered? 10 = Are the benefits worth the harms and costs? [63].

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
