# Peer review of "Where Are Smart Cities Heading? A Meta-Review and Guidelines for Future Research"

_applsci, doi:10.3390/app12168328_

Round 1
Reviewer 1 Report
The following observation is for the authors:
1. The contribution of the paper and the research GAP must be addressed in the introduction session. Since your article is a review, you must precisely mention what other reviews articles in the similar topics to yours have been written and indicate where your article falls in relation to those earlier works.
2. The reader needs to be able to understand your review mindset by seeing a graphic representation of how you constructed your article.
3. What fundamental concept divides your material and method into four separate sessions? and explain how it connects to the articles you have reviewed. This needs to be precisely explained to the reader and clearly identified.
4. The material and methodology employed in the earlier papers is one of the most crucial review conclusions. You need to pay closer attention and learn more about the ideas and techniques that other people employ.
5. I don't believe it is necessary to present the materials and review methodology in the review article. You must restructure your article such that the reader can clearly see how the field of study you aim to present has developed.
6. Please persuade me that your review articles contain information that is relevant to me and other readers in addition to the article's summary. You must present it so that everyone may understand it and gain more knowledge after reading your article (graphically, framework, procedure etc.)
7. Your article's Appendix, which is supposed to be attached to the main text, contains its strongest material. However, you should definitely rewrite your article.
8. Please reorganize the discussion to be more focused on the research direction in this field as it is too broad for me to reach the comparison, determination, or direction of the field.
Good luck
Author Response
Please find the response in the attached report document, thanks.

Reviewer 2 Report
-good idea abs Report, in good study and present
-only rewrite the main abstract to more clearly
-and add more references from ISI
- The article should better describe the use of the CASP (Critical Appraisal Skills Program)
- It will also be interesting to justify the use of Scopus in comparison with other databases.
There are a number of articles in the literature that compare databases (WoS, Scopus, Google Scholar) and that allow that justification. I suggest a search on Google Scholar in order to consider consulting these articles.
- Again, in the context of justifying the authors' options, why was NVIVO 12 used in comparison with other software? Those arguments are also missing.
- I conclude with the suggestion of highlighting in the introduction section the use of meta-review for research design.
Author Response

(The authors gave the same response as above.)

Reviewer 3 Report
Please find my review as an attachment.
Good Day!

Author Response

(The authors gave the same response as above.)

Round 2
Reviewer 1 Report
Good luck